# A Novel Steganography-Based Pattern for Print Matter Anti-Counterfeiting by Smartphone Cameras

**DOI:** 10.3390/s22093394

**Published:** 2022-04-28

**Authors:** Hong Zheng, Chengzhuo Zhou, Xi Li, Zhongyuan Guo, Tianyu Wang

**Affiliations:** 1College of Artificial Intelligence, Nanchang Institute of Science and Technology, Nanchang 330108, China; lixi@wit.edu.cn; 2School of Electronic Information, Wuhan University, Wuhan 430072, China; 2017301200283@whu.edu.cn (C.Z.); guozhongyuan@whu.edu.cn (Z.G.); 2020102120046@whu.edu.cn (T.W.)

**Keywords:** two-dimensional code, anti-counterfeiting printing, steganography, anti-copying pattern, semi-fragile

## Abstract

Print matter authentication based on anti-counterfeiting techniques has received continuously increasing concern from academia and industry. However, the existing printing anti-counterfeiting solutions often have the defects of poor identification experience, high cost, or weak anti-counterfeiting ability, and cannot achieve pre-sale anti-counterfeiting. Therefore, a novel steganography-based pattern for print matter anti-counterfeiting by smartphone cameras is proposed in this study. Firstly, every pixel in the original binary message image (such as QR code) is replaced by a square pixel block with the same binary gray value of 0 or 255 (the first-level expansion). Secondly, the obtained image is encrypted based on the logistic chaotic sequence, and then scrambled by Arnold transform. Lastly, once again every pixel in the generated image is replaced with a square pixel block (the second-level expansion), the size and gray value of which can be set to control the semi-fragile ability to distinguish an originally printed pattern from its illegitimate copy. If the message extracted from the printed pattern through the inverse procedure is complete enough to decode and read, the pattern is assumed to be an original print. Experimental results verify the advancement and effectiveness of the proposed scheme in distinguishing the copied pattern.

## 1. Introduction

In recent years, with the rapid development of digital technology and computer networks, more and more counterfeits—such as fake and inferior products, trademarks, and banknotes—are entering our society and affecting our daily lives. The volume of counterfeit products amounts to up to 2.5% of world trade [1]. Counterfeits seriously damage our social and economic order. Traditional anti-counterfeiting solutions often have the defects of poor identification experience, high cost, or inadequate anti-counterfeiting ability, and cannot achieve pre-sale anti-counterfeiting. Printed marks such as 2D barcodes have been widely used for information acquisition, product traceability, and identification. However, the information encoded in a common 2D barcode is generally accessible to everyone because of its open codec, and can be easily reproduced by digital image I/O devices with high resolution, and it is difficult to distinguish the original printed 2D barcode from its copy [2]. Therefore, it is necessary to apply anti-counterfeiting techniques to print matter to serve for authentication cases.

The current print matter anti-counterfeiting technology can be divided into two types: active and passive. Active mode refers to deliberately inserting special authentication information into the original information, which is difficult to copy. Passive approaches focus on intrinsic characteristics of raw information that cannot be cloned, such as the physically unclonable features of print matter [3,4,5].

Active methods have been widely used in print anti-counterfeiting technology, which can be subdivided into several aspects:

(1) Apply specific techniques or print materials [6,7]: Illegal copy attacks on print matter can be effectively prevented by using specific printing processes and materials that are inaccessible to counterfeiters. For instance, trademarks printed with thermochromic ink can be transparent at room temperature, and will turn visible when temperature enters a special range and then disappear when the temperature returns to normal [8]. A 2D barcode printed with fluorescent ink only turns visible when exposed to ultraviolet rays to prevent counterfeiting. Additionally, particular printing process such as intaglio printing can also prevent the duplication of print matter by printing with a unique 3D structure [9].

(2) Digital watermarking technology: Inserting an imperceptible digital watermark to the print mark, the watermark carries relevant authentication information [10,11]. In anti-counterfeiting cases, the semi-fragile watermark is applied most commonly [12,13]. The semi-fragile watermark is widely used in tampering detection, as its semi-fragile nature allows the watermark to show robustness to attacks with relatively low intensity and obviously change under high-intensity attacks. The insertion process can be implemented by modifying particular coefficients in the DCT domain [14,15]. The watermark extracted from the original print matter is relatively complete, while a watermark extracted from its illegal copy is obviously incomplete. Thus, an authentication judgement based on the completeness of the watermark can be implemented.

(3) Generate an anti-copying pattern: The anti-copying pattern is a pattern that carries information. Unlike the traditional two-dimensional code composed of black and white large pixel blocks, the anti-duplication pattern often has more delicate features that are sensitive to illegal copying attacks [16,17,18,19]. Anti-copying patterns include patterns combined with a high density of small block units, patterns with a large gray-value range, colorful patterns, etc. By detecting the details of the pattern, we can distinguish an originally printed pattern from its copy [20].

Although trademarks printed with specific techniques or print materials have high security, this raises their cost. As a result, print matter authentication based on digital anti-counterfeiting techniques has received continuously increasing concern from academia and industry. Anti-counterfeiting patterns for product identification are mainly printed on the surface of product packaging together with common 2D barcodes such as QR codes. The patterns guarantee the authenticity of products, while the common QR codes provide the ID for traceability. The pattern and QR code can be acquired by a smartphone camera and then verified and tracked in the cloud sever.

In terms of applying anti-counterfeiting patterns for print matter anti-counterfeiting, many researchers have already put forward corresponding schemes. Tkachenko I. et al. [21,22] proposed a two-level QR code (2LQR code); unlike standard QR codes, 2LQR codes use specific texture patterns instead of black-and-white block structures to construct a private level for private information or authentication. The patterns are sensitive to the print-and-scan (P&S) process, thus preventing copying. Xie N. et al. [23] proposed a LCAC (low-cost anti-copying) 2D barcode, the anti-duplication effect of which is achieved by adding location-confidential authentication information to the original information in the process of generating a QR code. Picard J. et al. [24] placed anti-copying patterns around or inside 2D barcodes. The pattern, combined with tiny blocks in black and white with different density, degenerates after duplicating. Nguyen H.P. et al. [25] proposed CGN (clipping Gaussian noise), where texture and degradations caused by the counterfeiting process will change the statistical behaviors of the embedded microtexture.

At present, many existing anti-counterfeiting patterns require the pattern to be captured by specific devices, such as scanners with high resolution, and the discrimination ability is not revealed for patterns captured by smartphone cameras, thus limiting the application scenario. Additionally, some existing scheme can be time-consuming. Moreover, the most reliable method of forging is to scan and reprint the pattern [21,23,25], and there is relatively little research on high-definition photography and reprinting after revision.

Based on the encryption and scrambling methods in steganography, combined with the operation of expanding pixel points into pixel blocks, we propose a novel anti-counterfeiting pattern. We assume that the original print–capture process of the generated pattern is a legitimate channel, while the capture–print–capture process is regarded as an illegitimate channel. Pixel blocks of different sizes and gray values are used as basic information units, and the degree of degradation is different after the first print–capture and the second print–capture. Firstly, every pixel in the original binary message image is replaced by a square pixel block with the same binary gray value of 0 or 255 (the first-level expansion). Secondly, an image encryption based on a logistic chaotic sequence is implemented to the image, and the image is then scrambled using Arnold’s cat transform. Lastly, once again every pixel in the generated image is replaced with a square pixel block (the second-level expansion), the size and gray value of which can be set to regulate and control the semi-fragile ability to distinguish an originally printed pattern from its illegitimate copy. During decoding, the binary information is evaluated by calculating the average gray value of a block, thereby improving the robustness against local attacks. If the extracted message is still readable, the pattern is assumed to be a legitimate one captured through the original print–capture process. The experimental results show that the proposed pattern has good confidentiality to distinguish it from patterns copied by general methods, with a relatively small physical size and convenient procedure.

The structure of this paper is as follows: Section 2 introduces the related works. Section 3 describes the research methods, and Section 4 presents the experimental results and analysis. Section 5 discusses the resilience of the pattern to common attacks. Section 6 summarizes the paper.

## 2. Related Work

### 2.1. Image Encryption Technology Based on a Logistic Chaotic Map

Based on existing research, signal encryption technology based on chaos technology has strong theoretical and practical value. Its security comes from the ultra-long period of chaotic signal, quasi-randomness, and the sensitivity of chaotic systems to initial state and system parameters. Logistic chaotic mapping is widely used in the existing image encryption algorithms because of its high security and efficiency [26,27,28,29]. Logistic mapping is defined as follows:X_k+1_ = μ × X_k_ × (1 − X_k_)(1)
where k = 0, 1, 2, …, X_k_ ∈ (0, 1), and the bifurcation parameter μ ∈ (3.569946, 4].

If we set up the initial sequence value X_0_ = x_0_, bifurcation parameter µ = µ_0_, and number of iterations N, a chaotic sequence with length N can be constructed as X = {x_0_, x_1_, x_2_, …, x_N−1_}, and then X is binarized to obtain sequence Y = {y_k_|y_k_ = round (x_k_), x_k_ ∈ X}. If N is greater than or equal to n × n, n is the size of the image to be encrypted. The last n × n elements in the sequence Y are arranged in rows to obtain a binary matrix ***B*** of n × n size. The matrix is used as the encryption mask, and finally the encrypted image ***C*** can be obtained by XOR operation between the original image ***A*** and the mask ***B***; its pixels are:***C_i,j_* = *A_i,j_*****⊕ *B_i,j_*,**(2)
where **⊕** is the XOR operation.

The corresponding decryption inverse operation is:***A_i,j_* = *C_i,j_* ⊕*****B_i,j_***.(3)

### 2.2. Image Scrambling Technology Based on Arnold Transform

As a kind of information-hiding technology, image scrambling technology refers to adjusting the spatial position of each pixel according to certain rules to create a chaotic visual effect. Image scrambling technology can not only eliminate the correlation between adjacent pixels, but also scatter the attacks on the image to all positions of the image in the reverse process, thus enhancing the image’s robustness to attacks [30,31,32].

Arnold transform, also called Arnold’s cat map (ACM), was invented by Vladimir Arnold in 1960 [33]. Two-dimensional Arnold transform is the most commonly used in image scrambling. For an image of size N × N, the Arnold transform is defined as follows:(4)$$\left[\begin{array}{c} x^{\prime }\\ y^{\prime } \end{array}\right]\;=\left(\begin{array}{cc} 1 & 1\\ 1 & 2 \end{array}\right)\left[\begin{array}{c} x\\ y \end{array}\right]\;(\mathrm{mod}\;N)$$
where (*x*, *y*) and (*x*′, *y*′) represent the coordinates of a pixel in the image before transformation and the coordinates of the pixel in the image after transformation, respectively, while mod is the remainder operation.

The inverse transformation formula is as follows: (5)$$\left[\begin{array}{c} x\\ y \end{array}\right]\;=\left(\begin{array}{cc} -1 & 2\\ -1 & -2 \end{array}\right)\left[\begin{array}{c} x^{\prime }\\ y^{\prime } \end{array}\right]\;(\mathrm{mod}\;N)$$
where (*x*^′^, *y*^′^) and (*x*, *y*) represent the coordinates of a pixel in the scrambled image and the coordinates of the pixel in the reconstructed image, respectively.

## 3. Methods

### 3.1. Steganography-Based Pattern Generation Algorithm

On the basis of information-hiding theory, we propose a novel steganography-based mode by combining multilevel block-expansion operations and an average grayscale-based decision strategy. The pattern generation algorithm mainly consists of the following steps:

(1) First-level block expansion of the original message: First, a standard QR code image ***m*** with size a × a is generated to store the authentication information of a 2D barcode. Then, each single pixel in ***m*** is represented by b × b pixels with the same pixel value—expanding by block. Thus, an intermediate binary image ***M*_0_** with size (a × b) × (a × b) is constructed.

(2) Image encryption and scrambling: The initial sequence value x_0_ and branch parameter of logistic mapping μ are set as secret keys; the number of iterations N is set to be greater than or equal to (a × b) × (a × b); a particular encryption mask ***L*** based on a chaotic sequence is generated, and then a bitwise XOR operation on ***L*** and ***M*_0_** is performed to obtain the chaotic image ***M*_1_**; a scrambling operation based on Arnold transform is performed on ***M*_1_**, the scrambling operation is repeated T times, and the binary encrypted image ***M*_2_** with size (a × b) × (a × b) is created.

(3) Second-level block expansion of the chaotic image: In order to enhance the robustness of a single pixel in ***M*_2_** to printing attacks, each individual pixel in the chaotic image ***M*_2_** is represented by c × c pixels. Unlike the first-level block expansion, the gray value of the block with size c × c can be the same as or differ from the corresponding binary pixel. The gray value of the block can be 0 or 255, while the gray value of the binary pixel is also 0 or 255. Moreover, when gray value of the binary pixel is 0, the gray value of the block can be set to a different value, such as 50; similarly, the gray value of the block can be set to 200 when gray value of the binary pixel is 255. Furthermore, the gray value of a block can be set to a random value in a particular range depending on the corresponding binary pixel; for instance, a single pixel with a gray value of 0 is replaced by a block with a randomly generated gray value in the range 0–127, while a pixel with a gray value of 255 is replaced by a block with a randomly generated gray value in the range 128–255. Finally, the digital image ***M*** of an anti-counterfeiting pattern with size (a × b × c) × (a × b × c) is generated, and the pattern can be printed along with the 2D barcode.

The generation procedure of the steganography-based pattern is shown in Figure 1.

### 3.2. Steganography-Based Pattern Authentication Algorithm

The pattern authentication algorithm mainly consists of the following steps:

(1) Segmentation and correction: Assume that the digital image ***P*** is obtained by photography in a clear situation. ***P*** is segmented and corrected to obtain the corrected image ***S*_0_** with size (a × b × c) × (a × b × c), where a, b, and c are the same as the values used in pattern generation.

(2) First-level evaluation of the captured image: In the decryption process, an adaptive Otsu binarization is first applied to image ***S*_0_**, and then it is divided into blocks of size c × c to obtain (a × b) × (a × b) sub-blocks in total, and the average gray level of each sub-block is calculated. If the average gray level is above a preset threshold (usually take 255/2 = 127), then the gray value of the pixel at the corresponding position is set to 255; otherwise, it is set to 0. In this way, a binary image ***S*_1_** with size (a × b) × (a × b) is generated.

(3) Unscrambling and decryption: First, T-round Arnold unscrambling is performed on the image ***S*_1_** containing a secret message. The initial sequence value x_0_, the branch parameter of logistic mapping μ, and the number of iterations N are set the same as in the generation process to reconstruct the encryption mask ***L*** based on the chaotic sequence, and a bitwise XOR operation is performed on ***L*** and ***S*_1_** to obtain the roughly decrypted image ***S*_2_**, whose size is still (a × b) × (a × b).

(4) Second-level evaluation of the roughly decrypted image: Similar to the first-level evaluation process, image ***S*_2_** is again divided into blocks of size b × b to get a × a sub-blocks in total, and the average gray level of each sub-block is calculated. If the average gray level is above a preset threshold (usually 255/2 = 127), then the gray value of the pixel at the corresponding position is set to 255; otherwise, it is set to 0. Finally, a binary image ***S*** with size a × a is generated in this way. We assume ***S*** is the decrypted image that carries the authentication message.

(5) Standard QR code decoding: The extracted image ***S*** is then decoded using a standard QR code decoding algorithm. If the message in ***S*** is still readable, then we assume that the pattern was captured through an original print–capture channel; otherwise, the pattern was generated through an illegal copy channel.

The authentication procedure of the steganography-based pattern is shown in Figure 2.

### 3.3. Algorithm Analysis

#### 3.3.1. Robustness to Local Attack

This algorithm uses two-level blocks to control the robustness of the pattern. The size of the first-level block mainly affects the resistance of the pattern to various interference or local attacks, which comes from the combination of block expansion and spatial scrambling. The pixel block expanded by a single pixel in the original secret message contains a plurality of pixels representing the pixel information in the original secret message, which are scattered to each position of the image after scrambling. Due to the chaotic encryption operation before scrambling, there is no periodic visual scrambling effect, which greatly improves the security of the original information. When the generated image is subjected to various local attacks (such as local blur, stain, loss, etc.), although the affected pixels are in adjacent regions in space, they actually come from different pixel blocks before scrambling; thus, the impact of local attacks can be dispersed to a certain extent through unscrambling operations and evaluation based on average gray levels.

#### 3.3.2. Semi-Fragility to the Print–Capture Process

The steganography-based patterns are roughly generated after the original secret message is encrypted and scrambled by first-level block expansion, but their robustness against the original print–capture process and fragility to illegal capture–print–capture processes need to be improved. The performance is mainly regulated by the size of the second-level block. The larger the second-level block, the stronger the robustness. However, as the block size increases, so does the possibility of forgery through the capture–print–capture process. If the block size is too small, the robustness against the original print–capture process will be weakened, so it may not be possible to extract identifiable information from an original print. The gray value of the pixel block used to replace the original single pixel in the second-level block expansion operation also has a certain impact on the semi-vulnerability of the pattern. As with standard two-dimensional code, the gray values of 0 and 255 are used to transfer the binary information. While beneficial for recognition after printing, this also makes the image easier to counterfeit through illegal reproduction processes. 

## 4. Experimental Results and Analysis

### 4.1. Experimental Setup

To evaluate the performance of our scheme, a robustness test after the original print–capture process and a fragility test after the illegal capture–print–capture process were conducted for the proposed pattern. We adjusted the block size in the process of the two-level expansion and the gray value in the second-level expansion to generate a series of patterns. The anti-counterfeiting ability of these patterns was compared under equal conditions. A digital image obtained by taking a picture of a first-time-printed pattern with a mobile phone was taken as the original pattern. Illegal copy prints were produced by direct copying and high-definition photo printing. The production of counterfeit patterns is shown in Figure 3. After printing, patterns were also taken by mobile phones, which are collectively referred to as copied patterns. 

According to [8], anti-counterfeiting ID differs for each product, and one of the main forms is a 16-digit sequence. A 16-digit sequence was separately encoded into two images of the standard QR code version V2 with error correction levels H and M as the original binary message. For simplicity, patterns generated from QR codes with correction level H are collectively referred to as H patterns, and patterns generated from QR codes with correction level M are collectively referred to as M patterns. The size of the standard QR code image was 25 pixels × 25 pixels, and all of the generated digital patterns were printed at a resolution ratio of 600 dpi. In order to make the physical size of the printed image about 1 cm × 1 cm, we set the first-level block size to 3 × 3 and 4 × 4, and the second-level block size was set to 2 × 2, 3 × 3, and 4 × 4 to form 2 × 3 = 6 combinations. The combination in which both of the two-level blocks’ size was 4 × 4 was removed due to its excessive physical size, so there were five combinations in total. In the second-level block expansion, with binary gray values of 0 and 255, two ranges of 0–110 and 140–255, and 0–50 and 200–255 were selected to represent the binary value, forming three combinations. In this way, a total of 5 × 3 = 15 digital images could be formed from the same basic secret message image, and there were 2 × 15 = 30 digital pattern images in total for the two basic secret message images. In order to improve the recognition rate, the corners of the extracted QR code were repaired.

To evaluate the recognition rates of different combinations, each printed pattern was photographed by a smartphone 50 times, so there were 50 × 30 = 1500 digital images in total for original printing. The recognition rate for each combination was obtained by dividing the number of instances of correct decoding by the total 50 attempts.

All of the experiments were implemented using Visual Studio 2017 with OpenCV 4.5.1 on MS Windows 10 Pro and an Intel Core i5-8250U (1.80 GHz) with 8.00 GB RAM. All of the original patterns and the copied patterns were printed on several sheets of coated paper using a Fuji Xerox Color C60 printer. A Huawei Nova 6 mobile phone with a 40-million-pixel camera was used as the end device for pattern capture. The magnification of the camera lens used when taking photos by mobile phone was 3.5.

### 4.2. Robustness Test after the Original Print–Capture Process

The block size in the process of the two-level expansion and the gray value in the second-level expansion were adjusted to generate a series of digital patterns. We decrypted the original print–captured digital image to extract a V2 QR code, and then decoded the QR code with the open decoder. The recognition results of the H patterns are shown in Table 1, and the recognition results of the M patterns are shown in Table 2.

As can be seen from the results, the robustness of the pattern to the original print–capture process is determined by the size of both of the two-level blocks. When the size of one level block is the same, the larger the size of the other level block, the stronger the robustness of the pattern. Comparing block combinations 3-4 and 4-3, it can be seen that the size of the second-level block has a greater impact on the robustness of the pattern, and the recognition results of patterns with different gray values also differ a lot. As expected, the binary image with gray values of 0 and 255 had the best recognition performance. Before decrypting the pattern, an adaptive Otsu binarization operation was implemented to better distinguish the two ranges. The error correction level of the original secret message also had an impact on pattern decryption and recognition, and the group of H patterns had a higher recognition rate.

### 4.3. Fragility Test after the Illegal Copy Process

#### 4.3.1. Direct Copy Forgery

The direct copy forgery method consists of scanning the primary print into a digital image through a scanner and then reprinting it to form a copy print. At present, many studies on scanning and printing processes show that the impact of scanning and printing processes on images is a combination of various forms of attacks, including spatial distortion—such as rotation, scaling, and clipping—and pixel distortion—such as gray diffusion—thus inevitably causing degradation.

We decrypted the digital image of the directly copied pattern to obtain a V2 QR code, and then decoded the QR code with the open decoder. The recognition results of the copied H patterns are shown in Table 3, and the recognition results of the copied M patterns are shown in Table 4.

As can be seen from the results, at this time, most of the extraction results of the combinations are unrecognizable, which can be considered as the loss of anti-counterfeiting information. Only those combinations with large block size and significantly different gray values (such as 0 and 255) can extract a still-recognizable QR code.

#### 4.3.2. HD Photographing and Printing Forgery

At present, the vulnerability of most anti-counterfeiting patterns is reflected in the loss of information in the anti-counterfeiting pattern to an unrecognizable extent for the secondary printed copies generated by direct copying and forgery. However, assuming that the attacker has some knowledge of image processing, they will capture images using high-definition devices such as smartphones, and apply some image processing algorithms before secondary printing, such as histogram equalization or binarization. These image processing technologies can improve the quality of forged patterns.

We used a smartphone to take photos at 3.5× magnification for an originally printed pattern to obtain an original print–captured digital image; then, after adaptive Otsu binarization, the binary image was printed again as copy of the original pattern. We decrypted the captured image of the HD photographed and printed pattern to obtain a V2 QR code, and then decoded the QR code with the open decoder. The recognition results of the HD copied H patterns are shown in Table 5, and the recognition results of the HD copied M patterns are shown in Table 6.

As can be seen from the results, similar to the former method of forgery, most of the recognition rates of the combinations are low, and only those combinations with large block size and significantly different gray values can extract a still-recognizable QR code. Compared with the directly copied test results, there are more nonzero recognition rates among the combinations. 

### 4.4. Time Consumption Test for Patterns of Different Size

The real-time ability of a system is vital for real-world situations. To evaluate the time complexity of the proposed pattern decrypting algorithm, a runtime test was conducted for patterns of different size. All 1500 of the original print–captured patterns were divided according to block size into five groups of 3-2, 3-3, 3-4, 4-2, and 4-3; each test set included 300 patterns. Average runtime was calculated by dividing the overall time consumption of pattern decryption by the total 300 attempts. The average runtime of each step for each group is shown in Figure 4. The standard deviation and range of runtime for the total decryption procedure are shown in Table 7.

Figure 4 shows that the total runtime is mainly determined by the Arnold unscrambling and logistic decryption procedure, along with the first-level evaluation. For patterns with the same first-level size, the runtimes of the unscramble and decrypt step are nearly the same, because the scramble and encrypt step is after the first-level expansion and before the second-level expansion in the generation procedure. Similarly, the second-level size mainly determines the runtime in the first-level evaluation. In general, parallel optimization towards iteration in the unscrambling and decryption procedure may help to reduce the runtime and improve the efficiency.

Stability is also important for real-time processing. As shown in Table 7, the decryption procedure has small runtime ranges, and the standard deviation for each group is around 1 ms, indicating the stability of the proposed decryption algorithm. 

### 4.5. Experimental Conclusion

Generally speaking, the robustness of the anti-counterfeiting pattern to the original print–capture process and its fragility to illegal copy processes are affected by many factors, such as the size of the two-level blocks, the value of the pixels, etc. It can be determined that the robustness to the first print–capture process and the fragility to the second print–capture process are usually contradictory. Therefore, it is necessary to select those combinations that can be correctly decrypted and decoded after the first print–capture process but cannot be correctly decrypted and decoded after the second print–capture process. Considering that the physical size should be as small as possible, and that using two gray ranges instead of a gray binary to represent the logical binary leads to better information-hiding ability, the QR code with error correction level H can be selected as the original secret message. Meanwhile, the first- and second-level block size is set to 3 and 2, respectively, and the gray-scale ranges of 0–110 and 140–255 are used to represent the logical binary values to form the pattern. The anti-counterfeiting ability of this combination is higher than that of the other combinations. With these parameters, the physical size of the printed pattern at 600 dpi is about 0.66 cm × 0.66 cm, which is the smallest among these combinations, thus entailing a lower printing cost. The digital H pattern and the original message’s QR code are shown in Figure 5. The original print–captured pattern and its decryption result are shown in Figure 6. The direct copy forgery pattern and its decryption result are shown in Figure 7, while the pattern generated by HD photography and printing and its decryption result are shown in Figure 8. It can be seen that the QR code extracted from the original print–captured pattern is relatively intact, and can be decoded, while the QR code extracted from the illegally copied pattern is quite different from the original QR code, and cannot be decoded even if the corners are repaired.

## 5. Discussion

### 5.1. Resilience to Defacing

The printing marks on the surface of products can be defaced in real-world circulation, and it is necessary to test their extractability with the pattern stained or cropped. However, the printed patterns had to be intact enough for authentication in previous studies, and few defacing tests have been conducted.

For the best-performing combination selected above, we filled the center of the captured pattern image with black square blocks of different size. A defaced pattern image with the length of the square defaced region being 50% of the total pattern length is shown in Figure 9. We tested the recognition rate of the QR code extracted from the defaced image, and the results are shown in Table 8.

It can be seen from the table that our steganography-based pattern has fine resilience to defacing, the possible reason for which is explored in Section 3.

### 5.2. Resilience to Blurring

Unlike images captured with a scanner, images captured by handheld devices such as smartphone cameras are often inevitably blurred due to motion or defocus [34,35]; it is also necessary to test their extractability when the captured pattern image is blurred.

For the best-performing combination above, we added different degrees of Gaussian blur to the captured pattern image with the filter size 5 × 5. A blurred image is shown in Figure 10. The recognition rate results are shown in Table 9.

It can be seen from the results that when the image is vaguer, the recognition rate decreases faster. The main reason for this is that the binary massage is evaluated completely according to the average pixel value during decryption, and the interpixel interference in a blurred image affects the evaluation.

### 5.3. Evaluation of Comparative Performance

In addition to the discrimination ability, cost, application conditions, and a certain degree of robustness to attack are also important metrics in real-world scenarios. The comparative performance of the proposed method and a typical anti-counterfeiting pattern is shown in Table 10.

As we mentioned in the Introduction, in real-world circulation, the printed anti-counterfeiting pattern is sent to and verified in a cloud server, and the common method is to calculate the similarity of some properties between the print–captured pattern and a corresponding pattern pre-stored in the server [5,21,36]. Additionally, the storage cost increases a lot as the patterns increase. In our proposed method, the print–captured pattern is decrypted and decoded to a 16-digit sequence; the sequence is then compared with a pre-stored sequence to obtain the authentication result, and the storage space per sequence can be small compared with a pattern image. Moreover, physical size can also limit the application, because an oversized pattern on print matter may affect its appearance. Under the same print resolution (600 dpi), the proposed pattern measures around 1 cm × 1 cm, the two-level QR code measures 1.2 cm × 1.2 cm, and the LCAC 2D barcode measures 3.2 cm × 3.2 cm in physical size.

## 6. Conclusions

Two-dimensional codes are powerful and widely used, but they are easy to copy. Anti-counterfeiting patterns are one of the effective methods to achieve anti-counterfeiting protection of print matter, including two-dimensional codes. The robustness of the original print–capture process is an important index to measure the detection ability of anti-counterfeiting patterns, but a large part of the existing anti-counterfeiting patterns need to be captured by printing and scanning equipment, or by particular equipment, such as high-magnification microscopes. For images captured by smartphone cameras, the anti-counterfeiting effect can be reduced or even lost. Therefore, it is still challenging to study the semi-fragile anti-counterfeiting patterns captured by smartphone camera. In this paper, a steganography-based pattern for print matter anti-counterfeiting by smartphone cameras is proposed, which takes advantage of the fact that the pixel blocks in the image will be subject to different degrees of attack after primary printing acquisition and secondary printing acquisition. The binary image with a secret message is subjected to first-level block expansion and encryption scrambling, and then the semi-vulnerability of the final generated pattern to the original print–capture process and the illegal copy process is regulated through second-level block expansion, and the average gray value decision is applied to enhance the robustness of the image to local attacks. Through our experiments, we found that the combination with relatively better anti-counterfeiting ability had the smallest size. In this combination, the QR code with an error correction level H was selected as the original secret message, with the first- and second-level block size set to 3 and 2, respectively, while gray scale ranges of 0–110 and 140–255 were used to represent the logical binary values to form the pattern.

While the proposed pattern is fragile to secondary printing and can be obtained via smartphone camera for anti-counterfeiting discrimination, the robustness of the pattern generated by primary printing to the motion blur and defocus blur that often appear in smartphone photography is still limited. The correct decoding and recognition of the first print–captured pattern must be carried out when the captured image is clear. In future works, how to find and use the anti-blur features of the image to implement the correct evaluation of blurred pattern and improve the recognition rate will be a problem to be studied and solved.

## Figures and Tables

**Figure 1 sensors-22-03394-f001:**
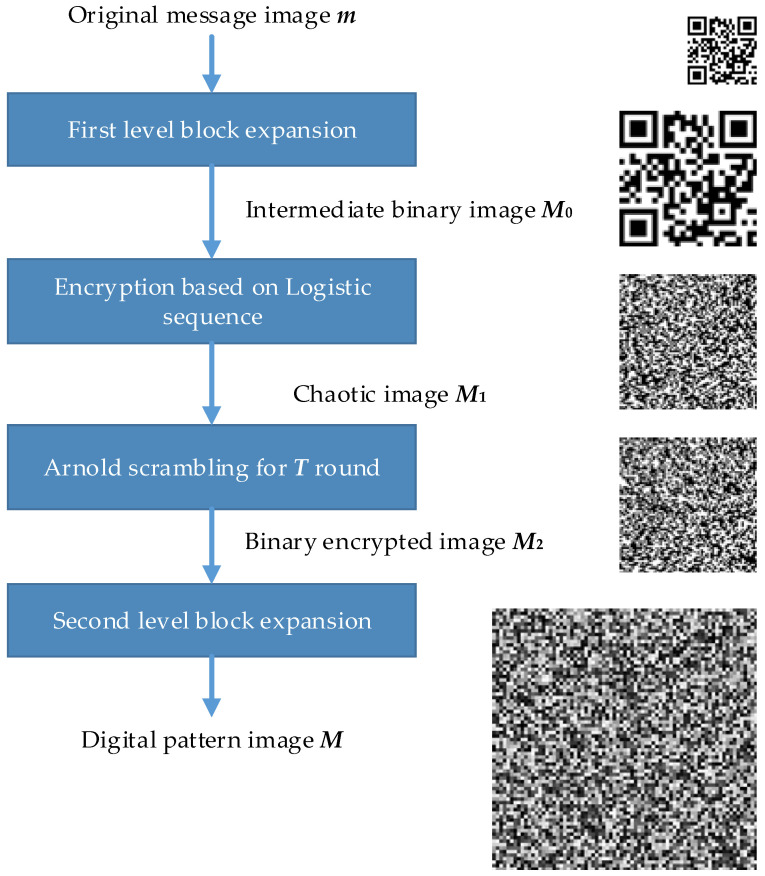
Generation procedure of the steganography-based pattern.

**Figure 2 sensors-22-03394-f002:**
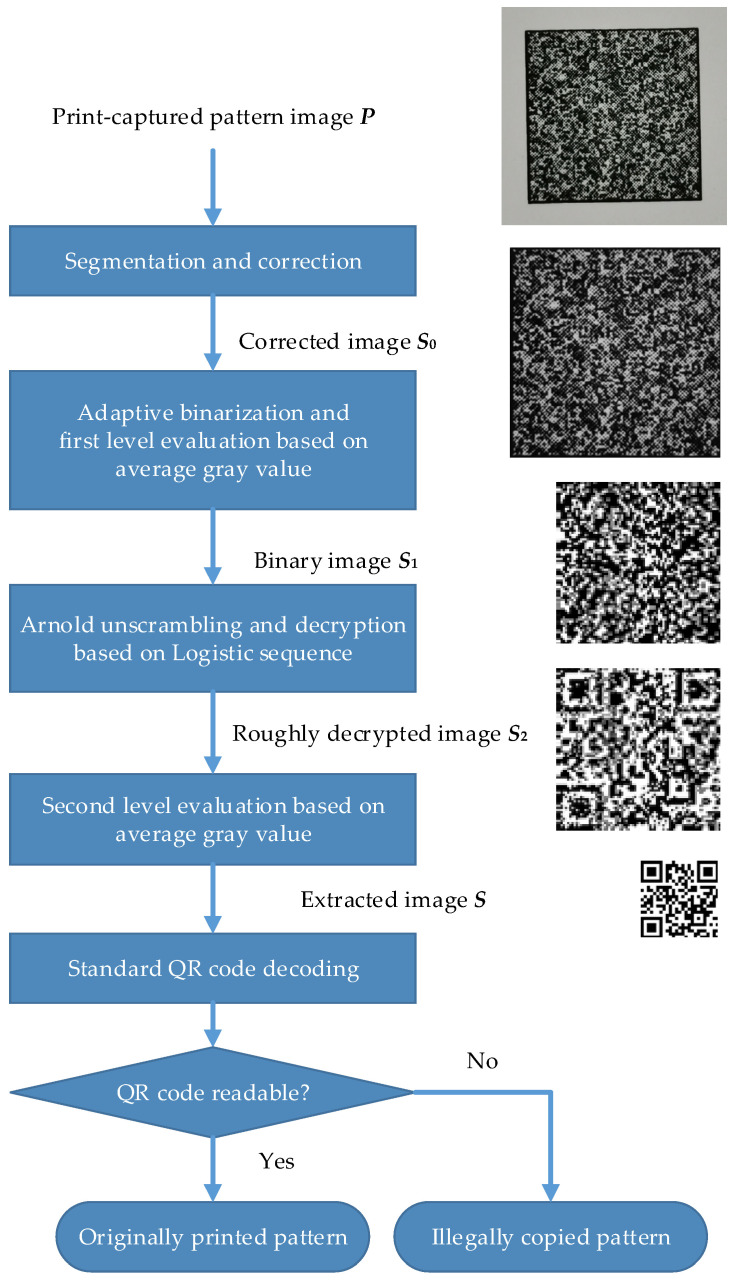
Authentication procedure of the steganography-based pattern.

**Figure 3 sensors-22-03394-f003:**
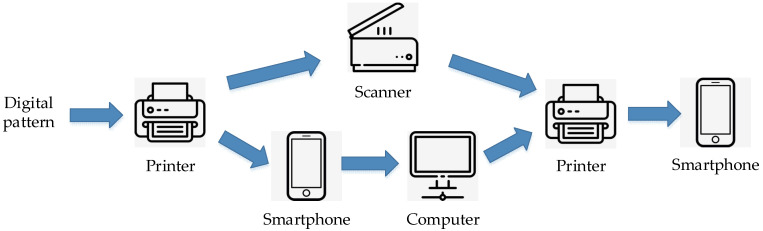
Production of the illegally copied pattern.

**Figure 4 sensors-22-03394-f004:**
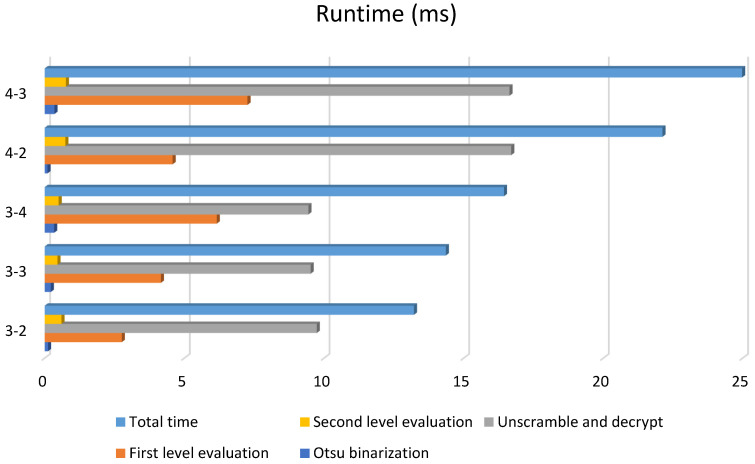
Average runtime for each group.

**Figure 5 sensors-22-03394-f005:**
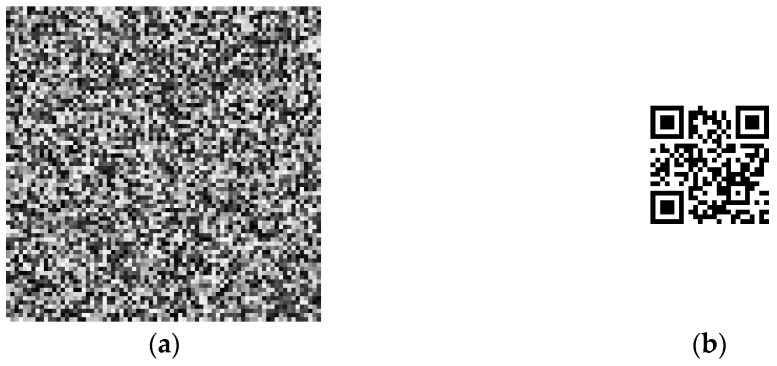
Digital pattern and original image: (**a**) digital pattern; (**b**) original image.

**Figure 6 sensors-22-03394-f006:**
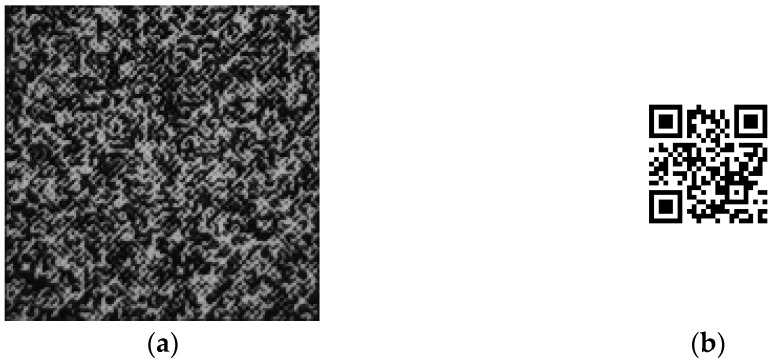
Original pattern and decrypted image: (**a**) captured pattern; (**b**) decrypted image.

**Figure 7 sensors-22-03394-f007:**
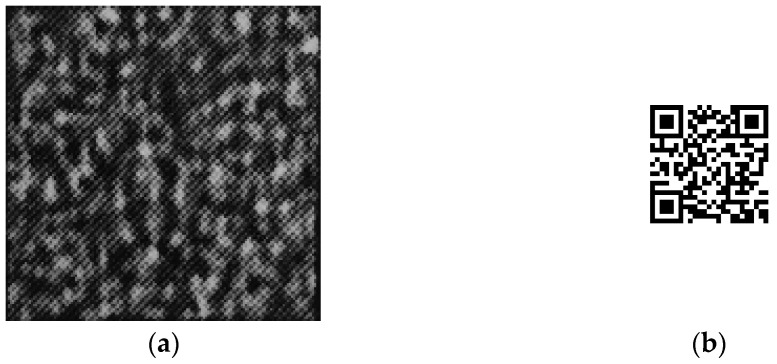
Copied pattern and decrypted image: (**a**) captured pattern; (**b**) decrypted image.

**Figure 8 sensors-22-03394-f008:**
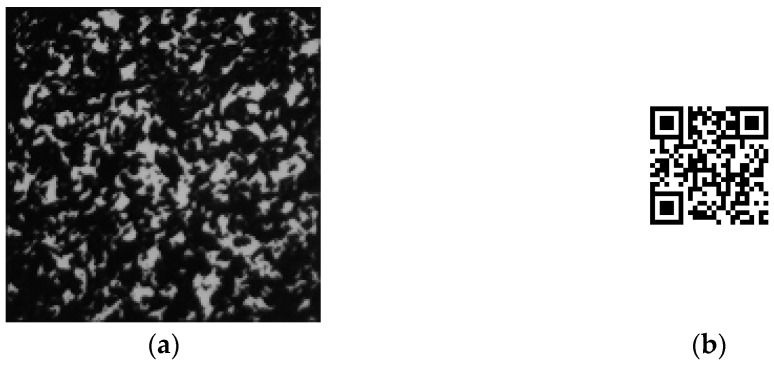
HD copied pattern and decrypted image: (**a**) captured pattern; (**b**) decrypted image.

**Figure 9 sensors-22-03394-f009:**
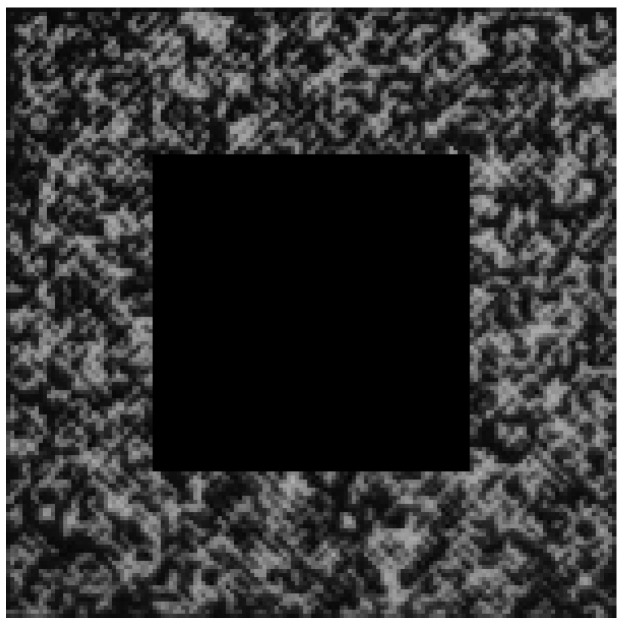
Defaced pattern.

**Figure 10 sensors-22-03394-f010:**
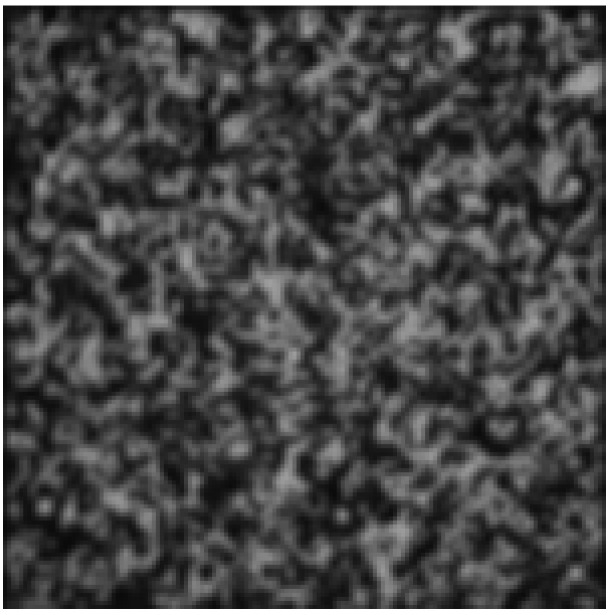
Blurred pattern.

**Table 1 sensors-22-03394-t001:** Recognition rate of the original print–captured H patterns.

	First–Second-Level Size	3-2	3-3	3-4	4-2	4-3
Gray Values or Ranges						
0 and 255		86%	98%	100%	96%	100%
0–50 and 200–255		86%	100%	100%	94%	100%
0–110 and 140–255		92%	98%	100%	88%	100%

**Table 2 sensors-22-03394-t002:** Recognition rate of the original print–captured M patterns.

	First–Second-Level Size	3-2	3-3	3-4	4-2	4-3
Gray Values or Ranges						
0 and 255		18%	94%	100%	88%	100%
0–50 and 200–255		44%	86%	100%	72%	100%
0–110 and 140–255		2%	60%	100%	50%	100%

**Table 3 sensors-22-03394-t003:** Recognition rates of the directly copied H patterns.

	First–Second-Level Size	3-2	3-3	3-4	4-2	4-3
Gray Values or Ranges						
0 and 255		0%	88%	100%	4%	98%
0–50 and 200–255		0%	68%	100%	0%	98%
0–110 and 140–255		0%	22%	96%	0%	88%

**Table 4 sensors-22-03394-t004:** Recognition rates of the directly copied M patterns.

	First–Second-Level Size	3-2	3-3	3-4	4-2	4-3
Gray Values or Ranges						
0 and 255		0%	0%	92%	0%	56%
0–50 and 200–255		0%	0%	88%	0%	2%
0–110 and 140–255		0%	0%	10%	0%	0%

**Table 5 sensors-22-03394-t005:** Recognition rates of the HD photographed and printed H patterns.

	First–Second-Level Size	3-2	3-3	3-4	4-2	4-3
Gray Values or Ranges						
0 and 255		0%	20%	56%	4%	52%
0–50 and 200–255		2%	40%	100%	12%	98%
0–110 and 140–255		0%	16%	94%	4%	60%

**Table 6 sensors-22-03394-t006:** Recognition rates of the HD photographed and printed M patterns.

	First–Second-Level Size	3-2	3-3	3-4	4-2	4-3
Gray Values or Ranges						
0 and 255		0%	8%	98%	0%	60%
0–50 and 200–255		0%	0%	68%	0%	34%
0–110 and 140–255		0%	2%	8%	0%	20%

**Table 7 sensors-22-03394-t007:** Standard deviation and range of runtime for the entire decryption procedure.

	3-2	3-3	3-4	4-2	4-3
Maximum time (ms)	17	19	20	23	32
Minimum time (ms)	12	14	15	21	24
Std dev (ms)	1.4060	1.1358	1.0809	0.5879	1.3500

**Table 8 sensors-22-03394-t008:** Recognition rate of defaced patterns.

Ratio of defaced region	0.2	0.3	0.4	0.5	0.6
Recognition rate (%)	88	84	68	40	0

**Table 9 sensors-22-03394-t009:** Recognition rate of blurred patterns.

Variance of Gaussian filter	0.5	1.0	1.25	1.5
Recognition rate (%)	86	60	34	4

**Table 10 sensors-22-03394-t010:** Comparative performance of different patterns.

Method	Captured by Smartphone Camera	Robustness to Defacing	Pattern Pre-Storage	Physical Size
Two-level QR code	No	Weak	Yes	Small
LCAC 2D barcode	Yes	Ordinary	No	Medium
Steganography-based pattern	Yes	Fine	No	Small

## Data Availability

Not applicable.

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
