# Peer review of "A Novel Steganography-Based Pattern for Print Matter Anti-Counterfeiting by Smartphone Cameras"

_sensors, 2022, doi:10.3390/s22093394_

Round 1

Reviewer 1 Report

In this paper, a novel steganography-based pattern for print matter anti-counterfeiting by smartphone cameras. The proposed method is based on taking advantage of the characteristics that the pixel blocks in the image will be subject to different degrees of attack after primary printing acquisition and secondary printing acquisition. In general, the proposed method is interesting and the simulation results seem to be promising. Unfortunately, the proposed mrthod is not clearly described and it is difficult for the reviewer to verify the correctness and to evaluate the value of the proposed method. I am a little bit concern about real implementation of the proposed method. However, I think complexity and delay analysis are missed in the work, it should be provided to enrich the paper. Authors are also invited to update references

Author Response

Dear reviewer,

Thank you for reviewing our manuscript and for the constructive comments, 
which greatly helped us to improve the manuscript. 
We have carefully taken your comments into consideration in preparing our revision.
The manuscript was carefully revised and point-by-point response was attached.

Reviewer 2 Report

While this is an interesting work indeed, I am having a hard time understanding the motivation behind it.

The introduction section does not clearly indicate why there is a real-world need for this proposition.

Furthermore, more recent references should be added (2018 - 2022). 

Isn't there a possibility whatsoever to carry out a comparison with counterpart schemes from the literature? If yes, then please provide it. If no, then this brings me again to the matter of the motivation behind this work.

Finally, a proof-reading of the manuscript would improve its quality. Very few grammar/spelling mistakes are found. An example is the spelling mistake in line 32.

Author Response

(The authors gave the same response as above.)

Reviewer 3 Report

  • Section "related works" is very poor. The related work includes only 4 references (24-27), and these studies are 10-12 years old published in proceedings (3-4 pages).
  • The captions and structure of Tables 2 and 3 are the same. It is not clear how they differ and what parameters are included inside and what accuracy measures are given. An analogous situation with the later tables (table 4 and 5 and etc.).
  • Table 1 is useless because nobody in the research paper gives the specification of the computer in the table.
  • Based on the experimental results, which range from 0 to 100%, it can be assumed that the solution does not guarantee stability.
  • Overall, the full text of the paper is very concise, with little information about the study, and the study's contribution to innovation is not revealed. Therefore, the scientific soundness is low,  it is not clear what the novelty and advantages are compared to the work of other authors

Author Response

(The authors gave the same response as above.)

Round 2

Reviewer 1 Report

The paper seems to be good, I thank the authors for their commitment.

Author Response

Dear reviewer,

Thank you very much for your evaluation and affirmation of our work.

Reviewer 3 Report

  • Please remove the first image ("Figure 1: Illustrative flowchart of the application of anti-counterfeiting models") as the diagram is too simple to need graphical representation.
  • More details on figure 5 are needed: how many iterations were used to calculate the means, what are the minimum and maximum values and what is the standard deviation? Why does the figure caption say per paterrn, but the text indicates that the average is for all 300 paterns?
  • the English needs to be revised. For example, please find another word for "decision". Perhaps the word "evaluation" could be used instead of "judgment"?

Author Response

Dear reviewer,

Thank you again for reviewing our manuscript and for the constructive comments, which greatly helped us to improve the manuscript. 
We have carefully taken your comments into consideration in preparing our revision.
The manuscript was carefully revised and point-by-point response was attached. 
